# Does Aging Affect Vitamin C Status Relative to Intake? Findings from NHANES 2017–2018

**DOI:** 10.3390/nu15040892

**Published:** 2023-02-10

**Authors:** Anitra C. Carr, Jens Lykkesfeldt

**Affiliations:** 1Nutrition in Medicine Research Group, Department of Pathology and Biomedical Science, University of Otago, Christchurch 8011, New Zealand; 2Faculty of Health & Medical Sciences, University of Copenhagen, DK-1870 Frederiksberg C, Denmark

**Keywords:** vitamin C, ascorbic acid, vitamin C requirements, vitamin C recommendations, aging, NHANES

## Abstract

The aging population is growing and fueling a global increase in chronic diseases and healthcare expenditure. In this study, we examine vitamin C dose–concentration relationships based on data from the National Health and Nutrition Examination Survey (NHANES) 2017–2018 to identify a possible age-dependent change in intake vs. concentration relationship among non-supplemented individuals (*n* = 2828). The vitamin C intake was similar between the younger (18–36 years), middle (37–58 years) and older (59–80+ years) age groups; however, circulating vitamin C concentrations were significantly lower in the middle and older age groups (*p* < 0.001). For intakes above 75 mg/day, no significant difference in the intake vs. serum concentration relationship was identified between younger and older individuals. However, for intakes below 75 mg/day, we found significantly lower serum concentrations relative to intake for the older compared to younger individuals, despite smoking being more prevalent in the younger compared to older adults (*p* < 0.001). This effect persisted among non-smokers and was further exacerbated by smoking in older people. Collectively, the present study suggests that healthy aging in non-institutionalized individuals does not increase requirements for vitamin C. In contrast, the lower serum concentrations relative to intake observed in older individuals at intakes < 75 mg/day may suggest that older individuals are more sensitive to a low vitamin C intake, perhaps due to the increased impact of long-term smoking and increased chronic disease prevalence in older adults. This finding may have implications for future intake guidelines in countries with low RDAs and for WHO/FAO, but requires further investigation.

## 1. Introduction

Vitamin C homeostasis is tightly controlled by a range of enzymatic and other processes governing its absorption, distribution, metabolism and excretion [1]. It has long been considered that these processes are both complex and dose-dependent, but as the maintenance of bodily processes in general becomes increasingly compromised with age, it has been speculated that vitamin C status may decline with age [2]. An age-dependent decline of vitamin C homeostasis could be due to altered pharmacokinetics, e.g., through declining absorption capacity, decreasing renal reabsorption capacity, increased renal leakage, impaired cellular energy balance, increased intracellular turnover from oxidative stress or simply due to a lower daily intake of the vitamin in older individuals. Some studies have indeed shown lower plasma concentrations in older compared to younger individuals, but more recently, conflicting evidence has suggested little or no difference with age, particularly in non-supplementing individuals [3].

In the US, the current recommended daily intakes (RDA) of vitamin C are 75 mg/day for women and 90 mg/day for men [4]. Moreover, an additional 10 mg/day is recommended in pregnancy, an additional 45 mg/day during lactation and an additional 35 mg/day for smokers, but there is no additional allowance for elderly. Similarly, the vast majority of health authorities have maintained a rationale based on the calculation of the requirements in adult men and then derived the assumed requirement for women and children using isometric scaling based on the relative body weight of a 70 kg man [5]. Thus, with one exception, previously identified differences between younger and older adults have not translated into health authorities publishing separate recommendations for vitamin C for older age groups. However, in their recommendation from 2001, a 10% increased reference value of 120 mg/day was set by the French authorities (Agence Française de Sécurité Sanitaire des Aliments, AFSSA) for adults aged 75 years and older, based on considerations related to immunity, cardiovascular risk, cancer risk and cognition [6,7].

With an increasingly aging population driving the global increase in chronic diseases and healthcare expenditure [8], the further investigation of modifiable determinants is warranted. In the present study, we examine vitamin C dose–concentration relationships based on data from the National Health and Nutrition Examination Survey (NHANES) 2017–2018 to determine if there is an age-dependent decline in the daily dietary intake or serum status of vitamin C or changes in the intake vs. concentration relationship.

## 2. Materials and Methods

### 2.1. NHANES 2017–2018 Cohort

Data from NHANES 2017–2018, available online from the National Center for Health Statistics (NCHS), the Centers for Disease Control and Prevention (CDC), were utilized for this study. NHANES is a nationally representative survey that uses a complex multistage probability sample to create a representative sample of the noninstitutionalized civilian US population. All participants provided informed consent and all identifying information was removed prior to the datasets being made publicly available online [9]. For the current analyses, inclusion criteria consisted of both sexes and all ethnicities, ages ≥ 18 years of age, of non-institutionalized civilian participants, who were able to provide informed consent, and participated in both questionnaire and laboratory measurements. From an initial dataset of *n* = 7435 with laboratory variables available, the following groups were excluded: those who were younger than 18 years, those with missing serum vitamin C values, those who were supplementing and those with no day 1 and/or day 2 dietary vitamin C intake data, resulting in a final cohort of *n* = 2828.

### 2.2. Demographic and Health Data

The following demographic information was extracted: sex (male or female), age (18 to 80+ years), ethnicity (Non-Hispanic White, Non-Hispanic Black, Mexican American, Other Hispanic, Non-Hispanic Asian or Other/Multirace), weight and body mass index (BMI). Smoking status (use of tobacco/nicotine in the last 5 days) and number of cigarettes smoked per day was also extracted, as well as the age the participant first started smoking cigarettes regularly in order to calculate the number of years of smoking. The prevalence of diagnosed health conditions was also extracted; these included various cardiovascular conditions (e.g., congestive heart failure, coronary heart disease, angina pectoris, myocardial infarction, stroke and hypertension), various lung conditions (e.g., emphysema, bronchitis and COPD), cancer, diabetes and other conditions (e.g., arthritis and gout).

### 2.3. Vitamin C Dietary Intake Data

Vitamin C intakes were captured using the Dietary Data Questionnaire dataset utilizing the What We Eat In America Questionnaire developed by the USA Department of Agriculture and USA Department of Health and Human Services. A total of 2 days of 24 h dietary recall data were collected through an initial in-person interview in the mobile examination clinic, and a second interview conducted over the telephone within 3 to 10 days. The USDA Food and Nutrient Database for Dietary Studies 2.0 (FNDDS 2.0) was utilized to determine the mean vitamin C intake. Vitamin C intake data are presented as mg/day.

### 2.4. Circulating Vitamin C Concentrations

Blood samples were collected from the participants by phlebotomists in the mobile examination clinic. Although the participants were not required to be fasting, the median (Q1, Q3) time of fasting was 10.5 (5.25, 12.75) h. The processed serum was immediately mixed with four parts 6% metaphosphoric acid and aliquoted in vials that were frozen at −70 °C. Vitamin C (ascorbic acid) was measured using isocratic ultra-high performance liquid chromatography (UPLC) with electrochemical detection [10]. Vitamin C concentration data are presented as µmol/L.

### 2.5. Data Analyses

Median and interquartile range (Q1, Q3) or mean and standard deviation (SD) were used for continuous variables and counts with percentages were used for categorical variables. Group differences were assessed using non-parametric Mann–Whitney U tests, with *p* < 0.05 signifying statistical significance. Sigmoidal (four parameter logistic) curves with asymmetrical 95% confidence intervals were fitted to dose–concentration data to estimate the vitamin C intakes required to reach ‘adequate’ serum vitamin C concentrations of 50 µmol/L and maximal serum concentrations achieved at intakes of 250 mg/day. Data analyses and graphical presentations were carried out using GraphPad Prism 9 (GraphPad, San Diego, CA, USA).

## 3. Results

### 3.1. Cohort Characteristics

The total non-supplementing cohort comprised 2828 participants (Table 1). The age range was from 18 to 80+ years with a median (Q1, Q3) age of 48 (32, 62) years. Of the total cohort, 50% were male and 25% had smoked in the last 5 days. One third of the cohort were non-Hispanic white and one quarter non-Hispanic black. The median body weight of the cohort was 80 (68, 97) kg and median BMI was 29 (25, 34) kg/m^2^. The cohort was divided into tertiles by age: 18–36 y (*n* = 942), 37–58 y (*n* = 942) and 59–80+ y (*n* = 944) for further analyses. The characteristics of the age tertiles are shown in Table 1.

### 3.2. Vitamin C Intake and Circulating Concentrations Relative to Age

The median vitamin C dietary intake of the total cohort was 53 (24, 102) mg/day, and was skewed due to some high intakes, resulting in a mean intake of 75 (95% CI 72, 78) mg/day. The median circulating vitamin C concentration of the cohort was 43 (23, 60) µmol/L, and was normally distributed. The age tertiles were assessed to determine if aging had an impact on vitamin C dietary intake and/or circulating concentrations. The older age group had a slightly higher vitamin C dietary intake (*p* = 0.03; Figure 1a), but lower circulating concentrations (*p* < 0.0001; Figure 1b) than the younger age group.

Males had lower median circulating vitamin C concentrations than females (39 {21, 55} µmol/L vs. 47 {27, 64} µmol/L, respectively; *p* < 0.0001), despite comparable dietary intakes to females (55 {24, 107} mg/d vs. 52 {25, 97} mg/d, respectively; *p* = 0.1). Within the age tertiles, there were no differences in vitamin C intake between the older and younger age groups for either males or females (*p* > 0.05; Figure 2a); however, circulating vitamin C concentrations were lower in the middle-aged and older relative to younger age group for both males and females (*p* <0.01; Figure 2b).

### 3.3. Vitamin C Dose–Concentration Relationship Relative to Age

Vitamin C status was assessed relative to dietary intake in the younger and older age groups (Figure 3). There did not appear to be a significant difference between the younger and older age groups with regard to intakes required to reach ‘adequate’ serum concentrations of 50 µmol/L (i.e., 67 {53, 82} mg/d vs. 93 {78, 112} mg/day, respectively) and maximal concentrations reached at intakes of 250 mg/day (i.e., 59 {55, 63} µmol/L vs. 62 {57, 67} µmol/L, respectively). In contrast, at intakes < 75 mg/day, significantly lower serum concentrations relative to dietary intakes were observed in the older compared to younger age group (Figure 3). Of note, there was a significantly higher proportion of smokers in the <75 mg/day vs. >75 mg/day intake groups (28% vs. 18%, *p* = < 0.0001), although this was true for both the younger (30% vs. 21%, respectively, *p* = 0.002) and older (21% vs. 15%, respectively, *p* = 0.03) age groups.

### 3.4. Impact of Smoking on Age-Related Dose–Concentration Relationship

Because smoking is known to have an impact on vitamin C status and requirements [11], the cohort was divided into non-smokers and smokers and dose–concentration relationships between younger and older adults were compared (Figure 4). Surprisingly, there was little difference in the vitamin C dose–concentration relationships between younger smokers and non-smokers (Figure 4a). A non-significant difference in the intake to reach 50 µmol/L of about 30 mg/day was observed in young smokers compared to non-smokers, corresponding to the current US recommendation of 35 mg/day additional vitamin C among smokers. However, older smokers had significantly higher requirements for vitamin C than older non-smokers, with >183 mg/day required to reach ‘adequate’ vitamin C concentration of 50 µmol/L compared with 86 (73, 104) mg/day intake in older non-smokers to reach the same serum concentration (Figure 4b). Among the cigarette smokers, the older age group had not only smoked for significantly longer than the younger age group (46 {40, 51} years vs. 11 {6, 17} years, respectively, *p* < 0.0001), but had also smoked a larger number of cigarettes per day (10 {5, 20} vs. 5 {3, 10}, respectively, *p* = 0.0006). However, despite there being large differences in the dose–response relationships between younger and older smokers (Figure 4d), a significant difference persisted between younger and older non-smokers at intakes of <75 mg/day (Figure 4c).

### 3.5. Prevalence of Health Conditions in Lower and Higher Intake Groups Relative to Age

The difference in vitamin C dose–concentration relationships at intakes < 75 mg/day between the younger and older age groups in the total cohort, and in the non-smoking subgroup, could be due to a higher prevalence of chronic diseases in the older age group, particularly at vitamin C intakes < 75 mg/day. To investigate this further, we compared the prevalence of a range of chronic health conditions in the older and younger ages groups, including for <75 mg/day vs. >75 mg/day intake subgroups. Not surprisingly, there was a significantly higher proportion of older people with various chronic health conditions (e.g., various cardiovascular diseases, cancer and diabetes) relative to the younger age group (*p* < 0.0001 for each health condition). Furthermore, with regard to the <75 mg/day vs. >75 mg/day intake subgroups, in the total cohort, there were a significantly higher proportion of people with diabetes (15% vs. 11%, *p* = 0.000) and stroke (6% vs. 3%, *p* = 0.0004) in the lower intake group. Similarly, in the older age group, there was not only a higher incidence of chronic diseases overall, but in the <75 mg/day vs. >75 mg/day intake subgroups, there were also a significantly higher proportion of older people with diabetes (30% vs. 22%, *p* = 0.007) and stroke (12% vs. 5%, *p* < 0.0001) in the lower intake group. In contrast, in the younger age group, there was a lower incidence of chronic diseases overall, and no significant differences between the lower and higher intake groups for either diabetes (2% vs. 1.5%, *p* = 0.06) or stroke (1% vs. 1%, *p* = 0.7). Thus, the differences observed in the vitamin C dose–concentration relationship between younger and older people at intakes < 75 mg/day could be due to a combination of a higher prevalence of chronic health conditions and the larger impact that long-term smoking appears to have on the dose–concentration relationship in older people.

## 4. Discussion

In this study, we examined if a possible age-dependency exists in the intake vs. serum concentration relationship among adults not using supplements based on data from the NHANES 2017–2018. Except for the French health authorities, none of the published national or international recommendations for vitamin C intake have included a particular recommended daily intake for elderly individuals [5,6]. For intakes above 75 mg/day, no significant difference was identified between younger and older individuals in the present study. Thus, this result supports the view adopted by most health authorities around the world promoting the same recommended daily intake of vitamin C for adults and people older than 70 years. Interestingly, however, for intakes below 75 mg/day, we found a significantly lower concentration to dose relationship for older compared to younger individuals. This observation partly questions the early pharmacokinetic studies published by Blanchard and coworkers [12,13]. These authors found no significant difference with age in any of the measured pharmacokinetic variables, including several absorption and clearance kinetics, in a detailed depletion–repletion study. However, the participants were all active individuals in good health, and more importantly, were non-smokers. Although entry level vitamin C status was not reported in these studies, an earlier observational study in a comparable cohort indicated saturating plasma vitamin C concentrations (i.e., 78 and 93 µmol/L for younger and older participants, respectively) [14]; thus, their results are likely more representative of the >75 mg/day group of the present study rather than adults ingesting lower amounts of vitamin C.

Although no difference in body weight was observed between groups, a difference was found in BMI. This may be interpreted as a higher degree of ‘obesity’ among the elderly but induced by a lower height as body weight is unchanged. Consequently, the observed lower vitamin C status in older vs. young individuals is not due to a volumetric dilution, but one may speculate that a potential impact of this ‘increased obesity’ (although the difference is numerically very small) may be through the increased oxidative stress and low-grade inflammation typically observed in obesity, again pointing towards increased vulnerability with age.

In our study, the age-related difference in the dose–concentration relationship persisted among non-smokers and was even further exacerbated by smoking in older people, despite smoking being more prevalent in younger compared to older people (*p* < 0.001). Interestingly, the (non-significant) difference in intake required among young smokers vs. non-smokers to reach 50 µmol/L was about 30 mg/day, i.e., corresponding well to the additional 35 mg/day recommended to smokers by the US authorities. Among older smokers, however, the amount necessary to compensate for the smoking habit was significantly higher and constituted at least 180 mg/day. These findings suggest that older individuals may be more sensitive to low vitamin C intakes, perhaps due to increased disease risk and a higher impact of long-term smoking on the dose–concentration relationship in older people. Indeed, we observed that not only had the older age group been smoking for significantly longer, as expected, but they also smoked significantly more than the younger age group. Smoking per se is known to increase oxidative stress and thus the consumption of vitamin C [15]. Although smoking has also been reported to impact diet, resulting in a lower vitamin C intake [11], this particular effect was eliminated in the current analysis. 

Among older individuals with an intake below 75 mg/day, we found a significantly increased incidence of various chronic diseases (e.g., diabetes and stroke), as well as an increased prevalence of smoking, all factors known to be associated lower vitamin C intake and plasma status [16,17]. Furthermore, as circulating vitamin C concentrations correlate closely with tissue levels of the vitamin [18], this suggests that older people with lower intakes may have a depleted tissue status, which may impact on the vitamin’s ability to carry out its various cofactor functions [5].

The NHANES 2017–2018 data have recently been evaluated in a comparative study examining the changes in vitamin C status in the adult US population between the NHANES 2003–2006 and 2017–2018 [3]. The authors found that the vitamin C status of the US adult populational was essentially unchanged over this period. However, they also examined the difference between age groups and found a significantly lower vitamin C concentration in the >60 years age group in NHANES 2017–2018 compared to 2003–2006. In the 2017–2028 cohort, an increased vitamin C concentration was found in older women apparently primarily driven by the increased use of supplements in this group. The study did not analyze intake vs. serum concentration relationships. In the present study, we excluded supplement users and found significantly lower serum status in middle and older age groups compared to the younger age group for both men and women (*p* < 0.01).

Our findings of a lower intake to concentration relationship among older individuals consuming <75 mg/day may have particular implications for those authorities with low RDAs for vitamin C for the general population, including the United Kingdom (40 mg/day) and WHO/FAO, Australia and New Zealand (45 mg/day) [5]. Even when following the national or international recommendations of these health authorities, older adults may have increased risk of experiencing significantly lower plasma concentrations compared to younger adults with similar intakes. In contrast, it would appear from the present data that countries with recommendations higher than 75 mg/day need not increase this for healthy older people in the general population.

For unhealthy older people, however, the present finding may suggest increased caution. Studies have shown that a wide range of diseases are associated with a lower plasma status of vitamin C, presumably due to a higher turnover, thus increasing the daily intake required to sustain an adequate plasma concentration [19]. The present study indicates that this issue may be even more relevant for older individuals requiring yet higher intakes to compensate for a higher prevalence of chronic disease. However, further studies are necessary to judge the relative importance of age in relation to vitamin C requirement during disease.

This is a particularly important consideration for institutionalized elderly as numerous comparative studies have shown a negative impact of institutionalization on the vitamin C status of older people [20,21,22,23]. Institutionalization is associated with a higher prevalence of chronic health conditions, including dementia, which are associated with decreased vitamin C status [24]. Furthermore, in nearly all cases, institutionalized older people had significantly lower dietary intakes of vitamin C than their non-institutionalized counterparts [20,21,23]. Thus, institutionalized older people have been suggested to potentially benefit from additional vitamin C dietary intake or oral supplementation to restore their vitamin C status to adequate levels [23].

The present study does not allow for conclusions on the putative mechanisms underlying the apparent increased sensitivity to low vitamin C intake among older people. However, the reasons may be several. A lower absorption of vitamin C could potentially be expected in older adults with inflammatory comorbidities due to the negative impact of inflammatory cytokines on intestinal vitamin C transport. Additionally, the delayed or diminished reabsorption of vitamin C from the kidneys would explain the observed right-shifted dose–concentration curve. Moreover, increased mitochondrial membrane leakage with age could contribute to increased oxidative and inflammatory stress, leading to an increased metabolic turnover of the vitamin C pool. However, none of these possible causes have been specifically investigated in the elderly at this point.

## 5. Conclusions

Collectively, the present study suggests that healthy aging in the general population does not increase requirements for vitamin C. Thus, the dose vs. serum concentration relationship is similar in younger compared to older people ingesting >75 mg vitamin C per day. In contrast, significantly lower serum concentrations relative to intake were observed in older vs. young individuals at intakes < 75 mg/day—particularly among smokers—which may have implications for future guidelines in countries with low RDAs and for WHO/FAO. Further research is required to unravel the determinants of the lower serum response of older people at low dietary intakes.

## Figures and Tables

**Figure 1 nutrients-15-00892-f001:**
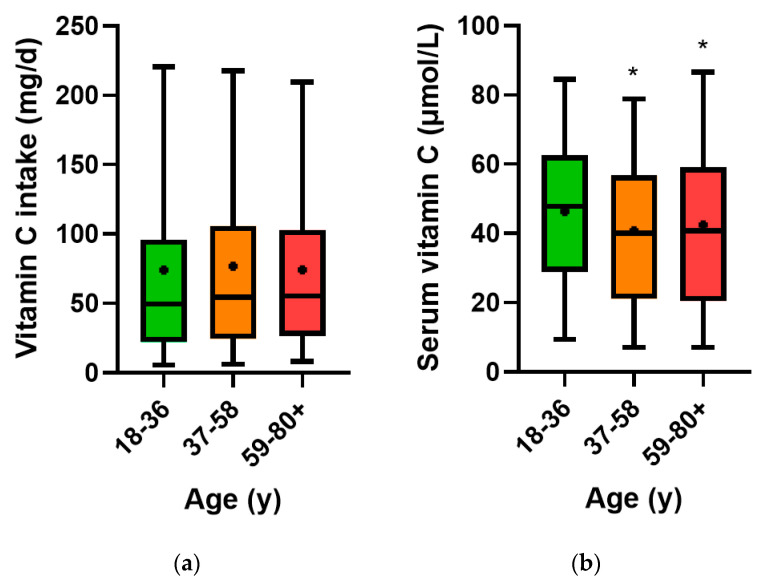
Dietary vitamin C intake (**a**) and circulating vitamin C concentrations (**b**) relative to age tertile. Age group: 18–36 (*n* = 942), 37–58 (*n* = 942) and 59–80+ (*n* = 944). Bars represent median with 25th and 75th percentiles as boundaries, whiskers represent 5th and 95th percentiles and symbols represent means. * *p* < 0.0001 relative to younger age group.

**Figure 2 nutrients-15-00892-f002:**
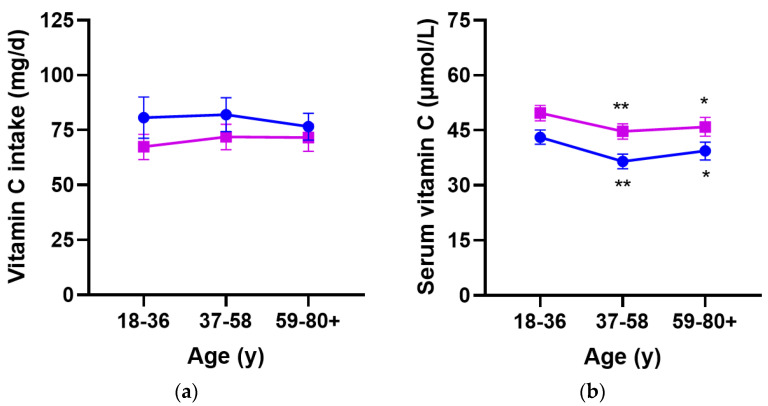
Dietary vitamin C intake (**a**) and circulating vitamin C concentrations (**b**) relative to gender within each age group. Blue circles represent males (*n* = 1425) and purple squares represent females (*n* = 1403). Data represent mean and 95% CI. * *p* < 0.01 or ** *p* < 0.001 relative to the younger age group.

**Figure 3 nutrients-15-00892-f003:**
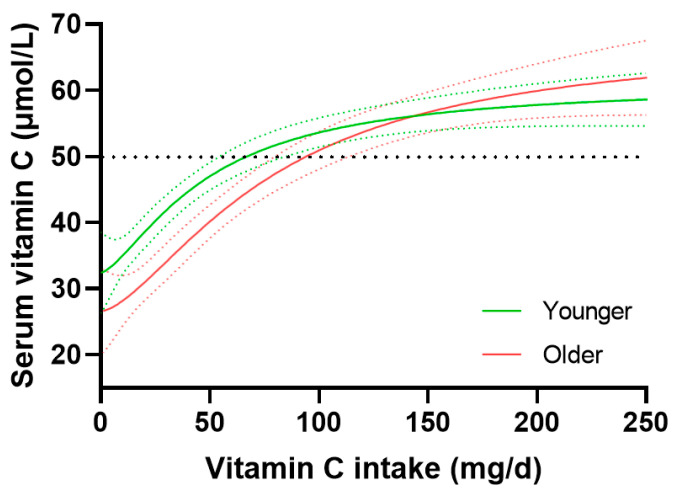
Circulating vitamin C concentrations relative to daily intake in the younger and older age groups. Younger age tertile of 18–36 y (*n* = 942) vs. older age tertile of 59–80+ y (*n* = 944). Sigmoidal (four parameter logistic) curves were fitted to the dose–concentration data with asymmetrical 95% confidence intervals indicated. Dashed line indicates 50 µmol/L serum vitamin C, which is considered ‘adequate’.

**Figure 4 nutrients-15-00892-f004:**
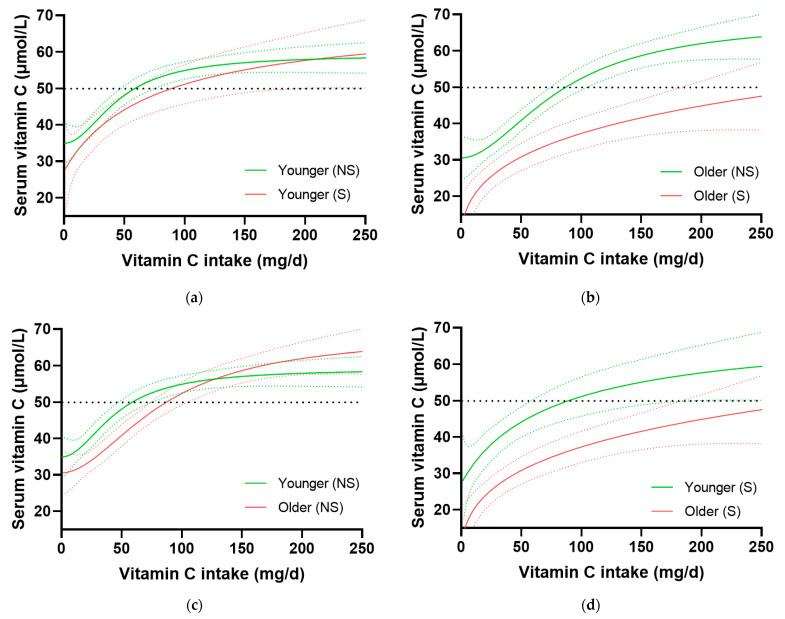
Impact of smoking on vitamin C dose–concentration relationships in younger and older age groups. (**a**) Younger non-smokers (*n* = 696) and smokers (*n* = 231). (**b**) Older non-smokers (*n* = 711) and smokers (*n* = 227). (**c**) Non-smokers (younger *n* = 696, older *n* = 711). (**d**) Smokers (younger *n* = 231, older *n* = 227). Sigmoidal (four parameter logistic) curves were fitted to the dose–concentration data with asymmetrical 95% confidence intervals indicated. Dashed line indicates 50 µmol/L serum vitamin C, which is considered ‘adequate’. NS, non-smokers; S, smokers.

**Table 1 nutrients-15-00892-t001:** Cohort characteristics relative to age tertiles.

Characteristics	Total Cohort (*n* = 2828)	Younger Age Group (*n* = 942)	Middle Age Group (*n* = 942)	Older Age Group (*n* = 944)	*p*-Value Y vs. O ^1^
Age, years:					
range	18–80+	18–36	37–58	59–80+	
median (Q1, Q3)	48 (32, 62)	26 (25, 32)	48 (42, 53)	66 (62, 73)	<0.0001
Sex, *n* (%):					
Male	1425 (50)	475 (50)	452 (48)	498 (53)	
Female	1402 (50)	467 (50)	490 (52)	446 (47)	0.3
Ethnicity:					
Non-Hispanic White	940 (33)	298 (32)	281 (30)	361 (38)	
Non-Hispanic Black	728 (26)	225 (24)	235 (25)	268 (28)	
Mexican American	399 (14)	143 (15)	149 (16)	107 (11)	
Non-Hispanic Asian	328 (12)	130 (14)	133 (14)	65 (7)	
Other Hispanic	281 (10)	85 (9)	89 (9)	107 (11)	
Other/Multi-race	152 (5)	61 (6)	55 (6)	36 (4)	0.04
Smoker ^2^	681 (25)	246 (27)	269 (29)	166 (18)	<0.0001
Body weight, kg	80 (68, 97)	79 (66, 96)	83 (70, 100)	80 (69, 95)	0.3
Body mass index, kg/m^2^	29 (25, 34)	28 (23, 34)	30 (26, 35)	29 (26,34)	<0.0001
Vitamin C intake, mg/d	53 (24, 102)	50 (22, 96)	54 (25, 106)	55 (27, 103)	0.03
Serum vitamin C, µmol/L	43 (23, 60)	48 (29, 63)	40 (21, 57)	41 (21, 49)	<0.0001

Data represent median (Q1, Q3) or *n* (%). ^1^ *p*-value is for Younger (Y) vs. Older (O) groups. ^2^ Data were missing for the smoking status of 79 (2.8%) participants.

## Data Availability

Data are publicly available from the Centers for Disease Control and Prevention’s National Center for Health Statistics: https://wwwn.cdc.gov/nchs/nhanes/cindex.htm.

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
