# Peer review of "Does Aging Affect Vitamin C Status Relative to Intake? Findings from NHANES 2017–2018"

_nutrients, 2023, doi:10.3390/nu15040892_

Round 1

Reviewer 1 Report

This manuscript is an excellent summary of findings from recent data collected as part of the NHANES studies (2017-2018).

The data collection is explained clearly and numbers are large enough to have confidence in the findings. The lower serum vitamin C levels despite greater intake and less smoking in the older population is a key finding, as is the potential for sex differences. The analysis showing that the impact of vitamin C depletion by smoking is greater in the aged group is particularly striking. It would be interesting to know if similar effects could be found for high oxidative stress chronic disease conditions (diabetes etc.), although such analysis may be beyond the capabilities of this data set. The authors do show that the incidence of these conditions is higher in the low intake group, although this is not quite the same as a comparison such as is performed for smokers

Although causes and mechanisms underlying these differences cannot be ascertained from this approach the authors are clear about any limitations to interpretation. These data should certainly be taken into consideration when trying to understand potential roles of age and chronic disease in determining healthy intake levels.

Minor point - In Figure 3, is there a reason that the Y-axis begins at 20 umol/L rather than 0. It may be slightly misleading to a non-conscientious observer.

Author Response

This manuscript is an excellent summary of findings from recent data collected as part of the NHANES studies (2017-2018).

The data collection is explained clearly and numbers are large enough to have confidence in the findings. The lower serum vitamin C levels despite greater intake and less smoking in the older population is a key finding, as is the potential for sex differences. The analysis showing that the impact of vitamin C depletion by smoking is greater in the aged group is particularly striking. It would be interesting to know if similar effects could be found for high oxidative stress chronic disease conditions (diabetes etc.), although such analysis may be beyond the capabilities of this data set. The authors do show that the incidence of these conditions is higher in the low intake group, although this is not quite the same as a comparison such as is performed for smokers.

COMMENT: The number of participants with the different diseases, particularly in the younger age group, makes carrying out the dose-concentration analyses untenable. Because the effect of smoking is overwhelming, it would be difficult to make the suggested analyses unless in non-smokers, which would further decrease the number of people with these diseases.

Although causes and mechanisms underlying these differences cannot be ascertained from this approach the authors are clear about any limitations to interpretation. These data should certainly be taken into consideration when trying to understand potential roles of age and chronic disease in determining healthy intake levels.

Minor point - In Figure 3, is there a reason that the Y-axis begins at 20 umol/L rather than 0. It may be slightly misleading to a non-conscientious observer.

ANSWER: Thank you for pointing this out – the y-axis has been reformatted to the same as the other figures i.e. it is now more obvious that it does not start at 0. We decided – for visual clarity of the effects – that it was better to start the y-axes at 20. We hope the referee can accept this.

Reviewer 2 Report

The investigators have interrogated the 2017-2018 NHANES database to identify a possible age-dependent change in intake vs serum concentration of vitamin C among 2,828 non-supplemented individuals. A major finding was that for intake below 75 mg/day they found significantly lower serum concentrations relative to intake in older (59-80+) than younger (18-36) individuals despite the increased incidence of smoke in younger folks. Importantly, the present study suggests that healthy aging in non-institutional individuals does not increase requirements for vitamin C. They also relate their findings to national and international recommended intakes of health authorities in various countries. The paper is well put forth and contains important nutritional information of interest to the public health nutritional community. 

This reviewer has only a few issues for the authors to consider in minor review. 

Comments 

  1. 1 Could the authors be a bit more specific as to the current US recommendations for vitamin C intakes?

  1. 2 Since vitamin E serum levels may in minor part relate to vitamin C are there any interrogations of vitamin E in the NHANES 2017-2018 data base? 

  1. 3 Since the elderly are “more obese”, do vitamin C serum levels reflect that condition? 

  1. 4 Should vitamin C serum intake levels be increased in active smokers? 

  1. 5 The journal discourages self citations. Although the reviewer recognizes the expertise of the authors in the field of vitamin C in the literature, the authors should consider consolidating some of their 10 self citations out of a total of 26 

Author Response

The investigators have interrogated the 2017-2018 NHANES database to identify a possible age-dependent change in intake vs serum concentration of vitamin C among 2,828 non-supplemented individuals. A major finding was that for intake below 75 mg/day they found significantly lower serum concentrations relative to intake in older (59-80+) than younger (18-36) individuals despite the increased incidence of smoke in younger folks. Importantly, the present study suggests that healthy aging in non-institutional individuals does not increase requirements for vitamin C. They also relate their findings to national and international recommended intakes of health authorities in various countries. The paper is well put forth and contains important nutritional information of interest to the public health nutritional community.  This reviewer has only a few issues for the authors to consider in minor review. 

Comments 

  1. Could the authors be a bit more specific as to the current US recommendations for vitamin C intakes?

ANSWER:  The current US recommendations have now been specified more clearly in the introduction.

  1. Since vitamin E serum levels may in minor part relate to vitamin C are there any interrogations of vitamin E in the NHANES 2017-2018 data base?

ANSWER: We agree with the referee that this would be interesting to explore but we have not been able to find any comparable vitamin E studies of the NHANES 2017-18 data.

  1. Since the elderly are “more obese”, do vitamin C serum levels reflect that condition?

ANSWER: This is an interesting idea. There is no difference in body weight between groups, so the ‘obesity’ difference is based on height. Consequently, this is not due to a volumetric dilution so a potential impact of ‘increased obesity’ (although the difference is numerically very small) may be through increased oxidative stress and low grade inflammation. This relevant point has now been added to the manuscript.

  1. Should vitamin C serum intake levels be increased in active smokers?

ANSWER: We assume the reviewer refers to an increased recommendation for smokers. The US and a few other countries already have additional recommendations for smokers although these may not fully compensate for the impact of the smoke. The present study found an additional intake of about 30 mg/d to be required for young smokers to achieve the same 50 µmol/L as young non-smokers (non-significant effect). However, a much larger and significant requirement was found for older smokers. This may suggest that the recommendation of 35 mg/d additional vitamin C for smokers is completely inadequate for older smokers. This has now been expanded in the discussion.

  1. The journal discourages self citations. Although the reviewer recognizes the expertise of the authors in the field of vitamin C in the literature, the authors should consider consolidating some of their 10 self citations out of a total of 26

ANSWER: The editor also noted their concern about self-citations, but this was related to citations to the journal rather than that of the authors. We have tried to take both concerns into account.

Reviewer 3 Report

This is a well-written and timely article from experts in the vitamin C field. This article updates the current knowledge on vitamin C levels and aging to readers. I have a few minor comments/suggestions to improve the quality of this manuscript.

Comments/suggestions:

1. Remove ageing from the keywords.

2. Can the authors discuss/speculate why there is decreased circulatory vitamin C concentration in the aged population? Is it due to reduced/decreased expression of vitamin C transporters in the absorptive/reabsorptive/other metabolically active tissues in the aged individuals?

Author Response

This is a well-written and timely article from experts in the vitamin C field. This article updates the current knowledge on vitamin C levels and aging to readers. I have a few minor comments/suggestions to improve the quality of this manuscript.

Comments/suggestions:

  1. Remove ageing from the keywords.

ANSWER: The keyword ‘ageing’ has been removed as suggested

  1. Can the authors discuss/speculate why there is decreased circulatory vitamin C concentration in the aged population? Is it due to reduced/decreased expression of vitamin C transporters in the absorptive/reabsorptive/other metabolically active tissues in the aged individuals?

ANSWER: This is indeed a possibility but it remains to be investigated. We have elaborated/speculated briefly on the possible underlying reasons in the discussion as requested.